# TAP: Efficient Derivation of Tensor Parallel Plans for Large Neural Networks

Ziji Shi*†, Le Jiang†, Jie Zhang†, Xianyan Jia†, Yong Li†, Chencan Wu†, Jialin Li*, Wei Lin†,

*National University of Singapore

†Alibaba Group

*Abstract*—**Model parallelism is essential to train large language models efficiently. However, determining the optimal model parallel schedule for a given neural network can be slow and inefficient due to the vast choice space. To address this challenge, we propose a tensor model parallelism framework called TAP, which automatically searches for the best data and tensor parallel schedules.**

**Our approach is based on the observation that a neural network can be represented as a directed acyclic graph, within which only exists a limited set of frequent subgraphs. With that, we design a graph pruning algorithm that efficiently folds the search space. As a result, TAP runs at sub-linear complexity with respect to model size, which makes it a practical solution for large-scale networks.**

**Experimental results demonstrate that TAP outperforms the state-of-the-art automatic parallelism frameworks by $20-160\times$ in searching time. Moreover, the performance of TAP's discovered schedules is competitive with expert-engineered ones. In summary, TAP provides a powerful and efficient tool for model parallelism that can help alleviate the burden of manual tuning.**

## I. INTRODUCTION

Recent years have witnessed a burgeoning of large deep neural networks (DNNs) that deliver unprecedented accuracy across a wide range of AI tasks. The rate of DNN model size increase, however, has far surpassed the growth in accelerator memory capacity. To address this challenge, model parallelism has been proposed, where model weights are sharded onto multiple devices during distributed DNN training.

There are two main paradigms in model parallelism: pipeline parallelism and tensor parallelism. Pipeline parallelism divides the model into layers. Only activations are communicated during the forward pass, while gradient tensors are exchanged in the backward phase. Pipeline parallelism has recently drawn much attention, with many proposed algorithms aiming to find the optimal pipeline schedule that minimizes the pipeline idle time (i.e., "bubble size"). However, pipeline parallelism suffers from two significant drawbacks: 1) each layer must fit into a single accelerator's memory, and 2) interleaving different layers can be challenging for models with imbalanced architectures. As an alternative, tensor parallelism partitions the model weights and distributes them to multiple devices, thus lifting the restriction on the size of individual layers. In this work, we focus on tensor parallelism.

Manual specification of tensor parallelism is a daunting task, given that the quality of a partitioning scheme depends on the neural network architecture and the hardware system. To address this challenge, *automatic* parallelism approaches have been proposed which leverage user hints or guided searches over the entire partitioning candidate space. We argue that a brute-force search of the space is unnecessary in the majority of cases. Our research makes two key observations: Firstly, most neural networks include shared subgraphs that can significantly reduce the search space. Secondly, communication is the primary bottleneck during tensor parallelism training, and contiguous partitions in a block cannot overlap. Therefore, the search process can be accelerated by only searching for unique neural network sub-modules and evaluating candidate strategies based on their communication cost.

Based on those observations, we present TAP , a deep learning framework that automatically derives tensor-parallel plans for arbitrary neural networks without requiring expert annotations. TAP first constructs a skimmed DAG by removing auxiliary nodes, then it finds all of the shared subgraphs and searches for the optimal sharding schedule for each of them. In the end, TAP reconstructs the DAG by applying the found solution to the original graph. TAP drastically reduces the search space for tensor parallel plans, achieving $20\times-160\times$ speedup compared with the state-of-the-art auto-parallel framework. Evaluations demonstrate that our approach can also generate comparable solutions to the tensor parallel schedules designed by an expert [17].

Our paper makes the following contributions:

- A set of intermediate representations (IRs) of the computational graph that abstract away from low-level implementation details;
- A graph pruning algorithm that exploits the shared substructure to facilitate efficient searching;
- A communication-based cost model that accurately captures the communication requirements for tensor-parallel training.

## II. BACKGROUND

### A. Model Parallelism

Model parallelism distributes model weights onto different devices and synchronizes the full model through collective communication [6]. Model parallelism can be further divided into categories, pipeline parallelism and tensor parallelism.

*1) Tensor Parallelism:* Tensor parallelism splits the model layer and distributes it across multiple devices, thus dispersing the computational overhead of the layer [17], [23], [26]. Each device stores only a portion of the input tensors in its local

memory. Therefore, the final result needs to be aggregated from partial results through collective communication. Tensor parallelism can alleviate the challenge of training heterogeneous models using pipeline parallelism and can achieve better performance.

### B. Automatic Parallelism

Automatic parallelism is a recent line of research on automatically distributing a local model from a single device to multiple devices using the data and model parallel strategies. Existing approaches for automatic parallelism rely on user hints or brute-force searches across the entire space.

*1) User hint:* User-hint-based automatic parallelism scales single-device programs to multi-device systems by incorporating user annotations. For instance, GSPMD [26] infers the operator partitioning scheme based on user annotations, while Whale [12] allows for the inclusion of user hints for semi-auto parallelization of large models and introduces a hardware-ware load balance algorithm. However, user-hint-based automatic parallelism approaches require users to possess a deep understanding of both the system and model, and hard-coded user hints may not be transferable when either the model or system changes.

*2) Search algorithm:* Recent work has proposed fully automatic approaches based on search algorithms to optimize distributed DNN training. For example, Tofu [25] uses a recursive search algorithm based on dynamic programming and DNN-specific heuristics to minimize communication for the entire dataflow graph. Flexflow [13] employs randomized search to find the best parallel strategy in the SOAP (Sample, Operator, Attribute, and Parameter) space. Alpa [28] optimizes large DL models through two-level optimizations: inter-operator and intra-operator. It enables inter-operator parallelism using dynamic programming and intra-operator parallelism with integer linear programming. Unity [24] represents parallelization and algebraic transformations as substitutions on a unified graph representation, uses a novel hierarchical search algorithm to identify an optimized sequence of graph substitutions, and scales to large numbers of GPUs and complex DNNs.

*3) Challenge of exploding search space:* Search-based approaches face the challenge of exploding search space as model size scales, resulting in significant time costs. For example, each tensor (assuming 2D) can be partitioned in three ways: not sharded, sharded along the first dimension (row-wise), or sharded along the second dimension (column-wise). Given a neural network $G(E, V)$ with $V$ weight tensors, there exists $3^V$ possible sharding plans. Therefore, finding an optimal sharding plan is an NP-hard problem.

## III. Approach

In this section, we formulate the problem of searching for an optimal tensor parallel schedule, followed by our observation of the common presence of shared sub-structures in a large neural network, leading to the motivation of our design.

### A. Problem Formulation

A neural network can be represented as a directed acyclic graph $G(E, V)$ comprised of $L$ layers. The set of vertices $V$ represents the operators, and the set of edges $E$ represents the data flow from producer to consumer operators. Operators can optionally carry a weight tensor. During the forward pass, an edge represents an activation tensor, while in the backward phase, it represents a gradient tensor. A layer $L_i \in L$ is either a layer or a cluster of operators with a similar composition. Let the physical training system be $S(m, n)$ where $m$ is the number of worker nodes, and $n$ is the number of accelerators per worker node. A parallel plan $p$ is a new graph mathematically equivalent to $G$. The cost function, $Cost(p, S)$, measures training latency for a given plan and training system. The goal is to find an optimal parallel plan $p^*$ where:

$$\underset{p}{\text{minimize}} \quad Cost(p, S)$$
$$\text{subject to} \quad p(X) = G(X) \forall X$$

How can an automated system find such a plan? Fig. 1 illustrates the typical workflow of an auto-parallel system. The system first reduces the search space for model splitting using pruning techniques. Next, a search algorithm is employed to generate one or more candidate plans for evaluation. Finally, a cost model evaluates all candidate plans and selects the one with the lowest cost based on predefined evaluation criteria.

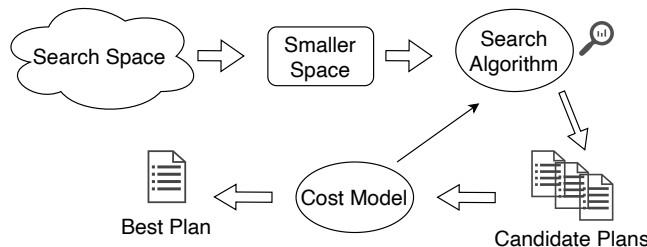

Fig. 1. General recipe of automatic model parallelism frameworks.

The end-to-end duration to produce an optimal schedule is a critical metric for an auto-parallel system. We identify three primary factors that contribute to the overall completion time: the size of the search space, the time complexity of the searching algorithm, and the speed of the evaluation method.

### B. Challenges and Observations

As mentioned earlier, a major challenge faced by auto-parallel systems is the search space explosion problem. This exponential increase in candidate space has led to impractical search time for modern large models [28] (§ V-B). This creates a dilemma: while auto-parallel systems aim to accelerate large model training, if *the derivation step itself* is too slow, it may offset the benefit of using an auto-parallel system.

How to effectively reduce this large candidate search space? To answer this question, we studied common scaling techniques for popular DNN models and summarized our findings in Table I. We observe that these techniques can be

| Scaling Technique | Task | Model | # Params | Shared Subgraph (SS) | # of SS |
|---|---|---|---|---|---|
| By width | Vision | ResNet50 [11] | 23M | Conv | 50× |
| | Vision + Language | CLIP-Base [18] | 63M | Transformer | 12× |
| | Language Model | WideNet [27] | 63M | MoE layer | 32× |
| | Vision | ViT-Huge [8] | 632M | Transformer | 32× |
| | Vision | V-MoE [22] | 15B | MoE layer | 24× |
| By depth | Speech | wav2vec 2.0 [3] | 317M | Conv, Transformer | 7×, 24× |
| | Language Model | BERT [7] | 340M | Transformer | 24× |
| | Language Model | T5-Large [19] | 770M | Transformer | 24× |
| | Language Model | GPT-3 [4] | 175B | Transformer | 96× |
| | Language Model | Switch Transformer [10] | 1571B | MoE layer | 15× |

TABLE I

SHARED SUBGRAPHS EXIST ON MANY NEURAL NETWORK MODELS. "CONV" MEANS CONVOLUTIONAL LAYER, "MOE" MEANS MIXTURE-OF-EXPERT LAYER.

grouped into two categories: scaling on the width, achieved by increasing the dimension of layers (e.g., adding more classes, attention heads, or convolutional channels), or scaling on the depth by increasing the number of layers. Notably, both techniques start with a *base subgraph*, a group of layers or operators, and expand from it. For instance, large pre-trained language models like BERT [7] and T5 [19] comprise tens of transformer layers, while multi-class object classification networks like ResNet-50 [11] are built from convolutional layers.

Furthermore, upon analyzing expert-designed parallel schedules ( [17], [20], [21]), we notice that *parallel schedules are predominately similar for layers of the same type*. This is due to the fact that similar layers have comparable computational and memory consumption. This finding motivates us to investigate *reusing* parallel schedules discovered for identical layers, which can reduce the search effort.

## IV. DESIGN AND IMPLEMENTATION

### A. Overview

Fig. 2 illustrates the workflow of TAP . Given a neural network represented as a graph, TAP first converts the graph into an intermediate representation(§ IV-B) called GraphNode and removes auxiliary nodes. TAP then performs graph pruning(§ IV-C) to restrict the search space from the complete graph to the subgraphs. After pruning, TAP explores the possible sharding opportunities using pre-defined sharding patterns(§ IV-D) and validates the candidate plans(§ IV-E). If a valid plan is found, it is evaluated using the cost model(§ IV-F). TAP takes the overall best plan, performs additional communication-level optimizations, and rewrites the model into a parallel version(§ IV-G). To use TAP , users only need to specify the device mesh as shown in the example below.

1. Example with TAP on 2 workers each with 8 GPUs

```
import tensor_auto_parallel as tap
mesh = [2, 8]
tap.auto_parallel(tap.split(mesh))
model_def()
```

### B. Intermediate Representation

TAP defines a family of high-level Intermediate Representations (IRs) to facilitate the derivation of parallel schedules. Compared to MLIR HLO [14], TAP IRs operate on a coarser granularity while preserving the necessary information for sharding.

Upon obtaining the original neural network graph, TAP first trims the graph by deleting the auxiliary operators (Step ① in Fig. 2). This will remove the initialization and checkpoint-related operators, which will be recovered when converted back to a neural network graph later. As a result, the remaining graph will consist of only computing and communication operators.

TAP IRs consists of:

*a) GraphNode.:* A GraphNode represents a group of computing or communication operators. It can be a layer or a logical group of operators, which is the basic unit for deriving the sharding schedule. The TAP graph is made of GraphNode while preserving the directed edges from the original DAG. Using the GraphNode IR, we reduce the number of nodes in the T5-large model from 60k to 1015 weight variables.

*b) Sharding pattern.:* A GraphNode could have multiple ways of sharding. For instance, a 2D matrix weight can be split on either dimension or replicated. TAP defines each sharding pattern using the SRC abstraction. TAP also establishes the cost of each sharding pattern based on communication cost.

*c) Sharding plan.:* A sharding plan is a set of subgraphs (blocks of GraphNodes) with sharding patterns connecting them.

### C. Pruning using Shared Subgraph

It is common for DNN models to contain shared subgraphs. If we could identify the shared subgraphs, we could prune the search space by searching only within the subgraph. We propose a graph pruning algorithm to compress the search space into a shared structure (Step ②):

In deep learning frameworks like TensorFlow [2], each variable is referred to by the operator that produces it. As such, variables under the same layer share the same name scope because they receive input from the same operator. Therefore, it is possible to cluster operators that fall under the same name scope.

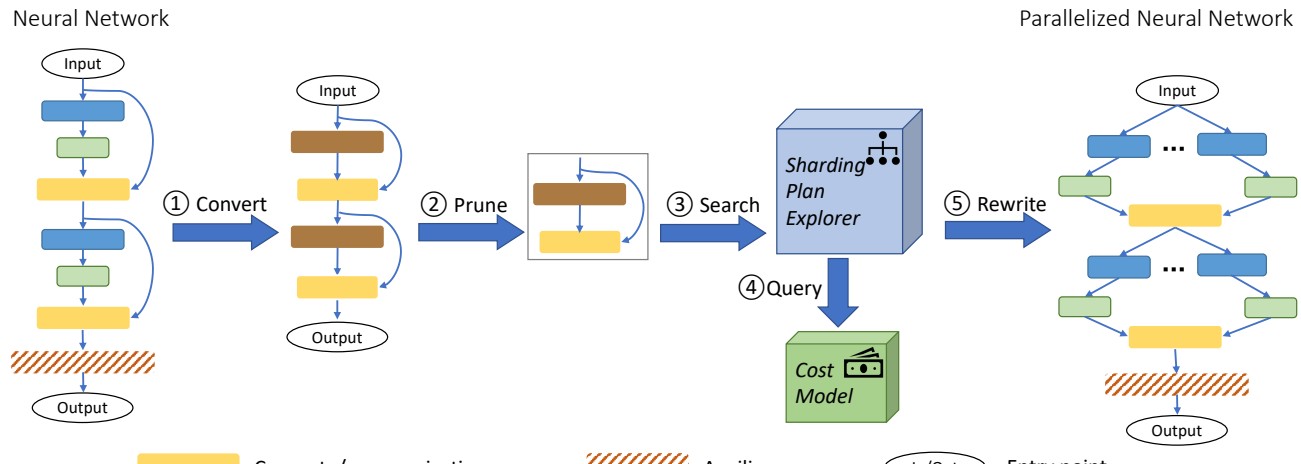

Fig. 2. Overview of the TAP system.

---

**Algorithm 1** Graph Pruning

1: **procedure** PRUNEGRAPH($modelDef, minDuplicate$)
2:     $nodeTree \leftarrow \emptyset$
3:     $maxDepth \leftarrow modelDef.depth$
4:     **for all** $depth \in maxDepth \cdots 1$ **do**
5:         $nodeTree[depth] \leftarrow$
    $longestCommonPrefix(modelDef.nodes.name)$
6:         $opCount = findSimilarBlk(nodeTree[depth])$
7:         **if** $opCount \geq minDuplicate$ **then**
8:             $subgraphs.append(nodeTree[depth])$
9:         **else**
10:           $break$
11:         **end if**
12:     **end for**
13:     **return** $subgraphs$
14: **end procedure**

---

**Algorithm 2** Derivation of Optimal Plan

1: **procedure** DERIVEPLAN($modelDef, shardingPatterns$)
2:     $subgraphs \leftarrow PruneGraph(modelDef)$
3:     $candidatePlans \leftarrow$
    $enumerateAllPlans(subgraphs)$
4:     $validPlans \leftarrow \{\}$
5:     **for all** $p \in candidatePlans$ **do**
6:         $validated \leftarrow PatternRouting(p)$
7:         **if** $validated$ **then**
8:             $validPlans.insert(p)$
9:         **end if**
10:     **end for**
11:     $bestPlan \leftarrow min(QueryCost(validPlans))$
12:     **return** $bestPlan$
13: **end procedure**

---

Algorithm 1 starts by constructing a *nodeTree*, which identifies and groups the GraphNodes on each level by using the longest common prefix algorithm on the GraphNodes names (line 2-5). After that, it finds the blocks of GraphNodes with a similar composition of operators and compares the number of operators with the minimum duplicate threshold (line 7). As the depth decreases, we will see a larger subgraph with less homogeneous compositions. Notice that multiple shared subgraphs may exist since a neural network may have multiple leaf nodes.

### D. Sharding Plan Generator

A sharding pattern, defining the way a GraphNode can be sharded, also serves as the directed edge between nodes. According to the SRC abstractions, the communication pattern is determined once the split/replica decision is made. Under the hood, the sharding patterns connect to each other like a chain.

After pruning, TAP proceeds to derive the optimal plan (Step ③ and ④) using Algorithm 2. In the first phase, TAP enumerates all possible sharding plans given the subgraphs. TAP only needs to work on hundreds of plans thanks to pruning. However, not every plan is valid because we only have weekly connected subgraphs. Therefore, the candidate plans need to be validated by checking the connectivity (line 5-10). Upon checking, TAP evaluates the performance of each plan using a cost model and selects the best plan.

### E. Pattern Routing

In the *pattern routing* step (Algorithm 3), TAP tries to assemble the weakly connected GraphNodes into a valid sharding plan by checking the connectivity. This is to ensure the success of graph rewriting (Step ⑤). TAP does so using breadth-first-search (BFS) starting from the root node, and the

**Algorithm 3** Plan Validation
```
 1: procedure PATTERNROUTING(currPlan)
 2:     TopoSort(currPlan)
 3:     nodesQ ← currPlan.root
 4:     while nodesQ ≠ ∅ do
 5:         currNode ← nodesQ.dequeue()
 6:         for all childNode ∈ currNode.next() do
 7:             sp ← lookUpShrdPatn(currNode, childNode)
 8:             if sp ≠ ∅ then
 9:                 if childNode == currPlan.leaf then
10:                     return TRUE
11:                 else
12:                     nodeQ.enqueue(childNode)
13:                 end if
14:             end if
15:         end for
16:     end while
17:     return FALSE
18: end procedure
```

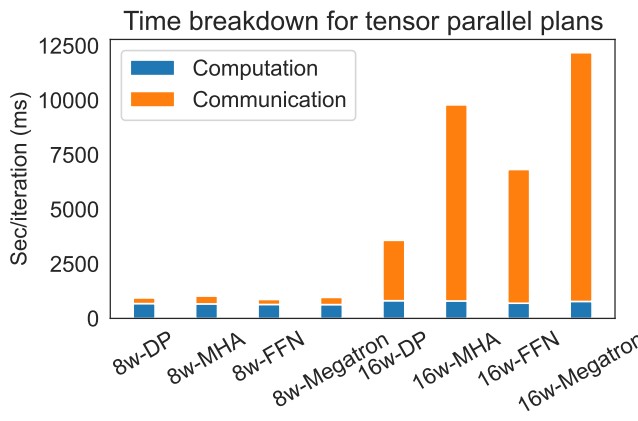

Fig. 3. Time breakdown for tensor parallel plans on T5-large model on 8 and 16 GPUs (8w/16w). DP means data parallel, MHA means sharding the multi-head attention, FFN means sharding the feed-forward layer, and Megatron refers to the tensor sharding plan described in [17].

goal is to make sure there exists at least a connected path from the root to the leaf chained using the sharding patterns.

One challenge is that a pair of contracting sharding patterns may have different input and output tensors, and a consumer operator's input is not ready until its producer is ready. In other words, dependencies exist between GraphNodes, but the information was kept in the original edges and could be lost when we perform pruning.

To solve it, we perform a topological search for the GraphNode based on the readiness of the input tensor. We leverage that neural networks can be represented using a directed acyclic graph, and reconstruct the edges based on the order of the producer-consumer relationship. This way, TAP avoids checking the order for every pair of GraphNodes.

### F. Cost Model

To build a cost model, we first profile different tensor parallel plans to understand the bottleneck. Fig. 3 summarizes the result. Data were collected from two nodes interconnected by 32 Gbps Ethernet, each equipped with 8 GPUs. We observe that *inter-node communication is the main bottleneck for tensor parallelism*, and *the best plan is not necessarily the one that splits every weight tensor*, in line with [6].

As the number of devices increases from $8\times$ to $16\times$, the difference between communication time and computation time is further pronounced. This is because the bottleneck has shifted from high-speed intra-node communication (PCI-e) to slower inter-node communication (Ethernet).

Furthermore, the best tensor parallel plan for 16 GPUs (*16w-FFN*) only shards the weight in the feed-forward layer. We conjecture that with more tensors split instead of replicated, there are fewer FLOPs per device and the computation time is lower. However, this comes at the cost of having more communication. In the case of training in the data center where nodes are interconnected by Ethernet, the speed bottleneck

may shift from computation to communication instead. Therefore, communication cost is the main consideration when we design the cost model.

TAP addresses these issues using an analytical cost model based on the tensor's communication method, shape, and data format. Each sharding pattern is associated with a cost, and the total cost is calculated by summing all pattern costs along the critical path.

### G. Graph Rewriting

After evaluating the cost of each sharding plan, TAP assembles the parallel plan. It does so by first restoring the original order of operators. Then, TAP identifies optimization opportunities that can be performed through gradient packing. In the end, TAP passes the resulting parallelized neural network plan to the deep learning framework runtime.

### H. Limitation and Future Work

To further optimize the memory consumption, TAP could leverage other orthogonal techniques such as Auto Mixed Precision (AMP) [1], recomputation [5], and pipeline parallelism. Since both AMP and TAP optimize on the graph representation of the neural network, they can be made into different passes. Also, gradient checkpointing can be used to offload the selected GraphNode onto the main memory. TAP may also be used with pipeline parallelism through automatic [9], [12], [15], [16] or manual placements.

## V. PRELIMINARY EVALUATION

### A. Setup

We first evaluate the pruning algorithm and the use of Just-In-Time compilation for TAP . Then, for comparison with another auto-parallel framework, we use Alpa version 0.7 running with JAX 0.3.5. Next, we use Megatron running on PyTorch to compare against expert-engineered tensor parallel

plans. Finally, we present the training convergence running gigantic neural networks.

The evaluation was performed on Company A's public cloud node with 756GB main memory, $2\times$ Intel 8163 CPUs at 24 cores each, and $8\times$ Nvidia V100 SXM2 32GB GPUs. Additionally, TAP builds on top of TensorFlow 1.12.

### B. End-to-End Evaluation

In this section, we compare TAP with auto-parallel framework Alpa on search time and performance of the discovered plan.

*1) Search time.:* As explained in § **??**, TAP has a sublinear time complexity, which is desirable when the models' size scales up. In the experiments with Alpa, we present the end-to-end search time with respect to model scaling, defined by the duration from the start of the experiment till the moment that the training process begins. Due to time constraints, we shortlisted a search space of 16 plans for T5 and 5 plans for ResNet, while we did not restrict the search space for TAP .

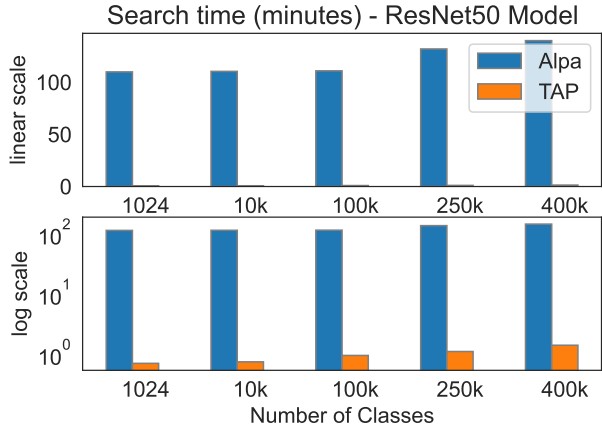

Fig. 5. End-to-end search time when scaling on the number of parameters for the large-scale classification model.

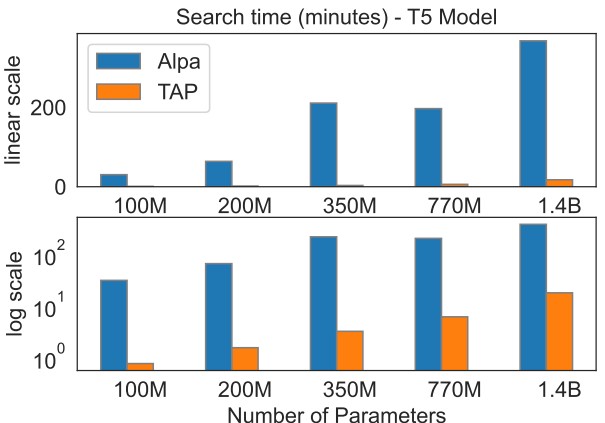

Fig. 4. End-to-end search time when scaling on the number of parameters for dense transformer model.

To scale the model along the depth, we increase the number of transformer layers for T5, an encoder-decoder transformer architecture for language modeling. Increasing the depth of dense transformer models is a common practice to improve performance. Fig. 4 shows that, with rising parameters, TAP can still find a plausible schedule in under 15 mins, which is $21\times - 67\times$ faster than Alpa.

To scale the model size along the width for the ResNet50 model, we choose to increase the size of the classification layer. The original ResNet50 model has 1024 classes in the classification layer. As we increase the dimensions for the classification layer, the total number of parameters also scales up. As shown in Fig. 5, TAP is two orders of magnitude faster than Alpa in finding the optimal solution. Our system outperforms it by $103\times - 162\times$.

We further analyze the time breakdown during the search. For example, for 24-layer T5-large (770M parameters), Alpa spent 5 mins profiling the operators and 5 mins constructing

the pipeline stages out of the operators. Instead, TAP reduces the architecture to one transformer block and searches for shardable parameters within that only, drastically reducing the search space. As a result, Alpa takes 197 minutes to search for 16 candidate plans, while TAP requires only 6 minutes to examine 729 candidate plans.

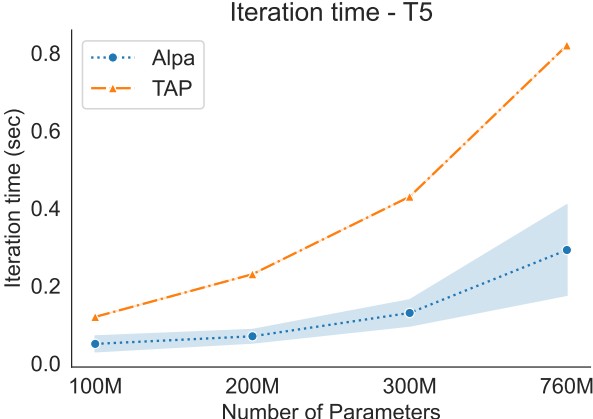

Fig. 6. Training time per iteration for T5 (batch size=16). The blue band represents the standard derivation.

*2) Training speed.:* We also evaluate the performance of the best plans produced by Alpa and TAP . We observe that Alpa favors pipeline parallel schedules, while the optimal schedule found by TAP is similar to the Megatron-style tensor parallel schedule. Since the plans using pipeline parallelism require less communication, the plans from Alpa have a higher throughput.

We also observe that as the width of the model increases, the performance of TAP plans is better and more consistent. Fig. 7 shows the time to finish one iteration of training for parallel plans of ResNet50. We first observe that TAP consistently

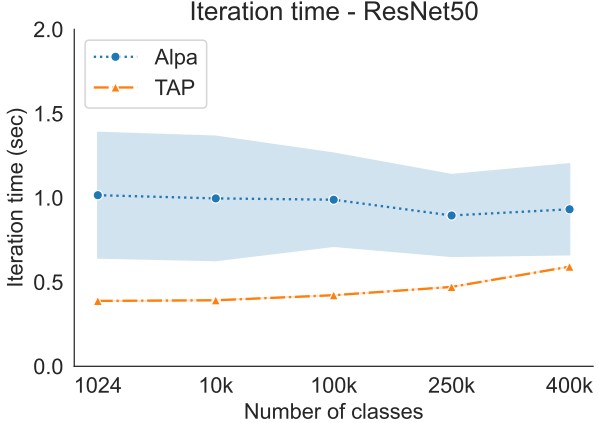

Fig. 7. Training time per iteration for ResNet50 (batch size=1024).

outperforms Alpa. Further, the variance (blue band) in plans discovered by Alpa shows that it struggles to find consistently good plans.

## VI. Conclusion

We present TAP, an automatic parallelism framework that efficiently discovers tensor parallel plans for large models. Leveraging the observation that shared subgraphs widely exist in neural networks, we design a pruning algorithm that efficiently reduces the search space with a sub-linear end-to-end complexity. The best plans found by TAP are comparable with the state-of-the-art expert-engineered plans while only taking minutes to discover.

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
