# OpenReview forum: "TAP: Efficient Derivation of Tensor Parallel Plans for Large Neural Networks"
_iscaconf.org/ISCA/2023/Workshop/ASSYST — ASSYST Oral_

### Official Review · Reviewer_yMJy · 2023-05-03
**The idea is neat, but the experiment results are not sufficiently convincing.**

**Rating:** 5
**Confidence:** 4

**Review:**

**Summary**
The author propose a search method to search in the design space of tensor parallelism. The results show 20-160x search time improvement over other works.




**Review (Strengths/Weaknesses):**

**Strengths**

* The idea is presented in a very clear way. The related works are well referenced. The authors are knowledgeable in this space.


**Weaknesses**
* The experiment results are the most concerning part for me.
1) Why Alpa is pick to be the baseline need to be stated clearly. Without knowing the quality of the baseline, it makes the comparison not convincing.
2) Is Fig.6 result suggesting Alpa is much better than TAP? If so, it would need some explanation.
3) Fig. 7 is also drawing my concern. It looks like TAP is not as scalable as Alpa, since Alpa is mostly flat and TAP time increases when scaling up.
4) If the experiment results are based on developed Cost Model. The validation of cost model need to be presented. Otherwise, it is hard to believe the results.
5) I would want to see the actual deployment result on at least one found schedule by Alpa and TAP.

* I like that the authors present TAP in detail in Sec. IV. However, I would appreciate if the author can have a high-level figure which guide the reader through all the three algorithms (Fig. 2 is vague and not really helpful). Also, I suggest before going step by step in each algorithm, providing some high-level summary about what each algorithm is doing would help the reader understand.


* [Minor] There are some typo and missing reference in the paper.

**Reviewer Expertise:**

Knowledgeable: I used to work in this area and/or I try to keep up with the literature but might not know the latest developments.

---

### Official Review · Reviewer_v2by · 2023-05-05
**Good improvements but not enough technical detail.**

**Rating:** 5
**Confidence:** 3

**Review:**

I enjoyed reading the paper, and found the approach of finding subgraphs in large neural network graphs and exploring parallelism schemes in these subgraphs to reduce the search space sensible. I was a bit disappointed not to see any detail on the GraphNode IR, especially since this is listed as a main contribution

**Review (Strengths/Weaknesses):**

Strengths:
* This is definitely an important problem.
* Well-written paper.
* Uses well-understood techniques such as graph pruning in a new domain of neural network parallelism exploration.
* 20-160X improvement in search time of parallel schedule finding.

Weaknesses:
* TAP is not able to reach the scalability of the baseline in T5.
* TAP is only evaluated on two models: T5 and ResNet50.
* TAP does not support pipeline parallelism, which partly explains why Alpa gives a better parallel partitioning scheme over TAP in T5 and also why T5 can be faster in search (ignoring pipeline parallelism shrinks the search space).
* The IR is claimed to be a contribution, but there is almost no detail about the IR in the paper.


**Reviewer Expertise:**

Knowledgeable: I used to work in this area and/or I try to keep up with the literature but might not know the latest developments.

---

### Official Review · Reviewer_9cDr · 2023-05-09
**Needs improvement in writing and explanation**

**Rating:** 5
**Confidence:** 2

**Review:**

The paper presents a search space pruning technique to explore the optimal parallel configurations for the distributed training of neural network model on multi-node datacenter-type systems. I have a few concerns:

1. Please check the grammar, spellings, and abbreviations in the entire paper. Abbreviations are used without being defined.

2. Fig. 6 is contradictory. Why is the cost model not defined to find an optimal schedule? Why is it defined to search on tensor parallel schedules. This seems like a half-baked work. Can this be made more generic? I feel that would be more desired by the community depending on what cost metric they want to optimize.

3. If the cost function is training latency, then in fig. 6 why is the iteration time for alpha better than TAP? It seems alpha is optimal here.



**Review (Strengths/Weaknesses):**

1. Please check the grammar, spellings, and abbreviations in the entire paper. Abbreviations are used without being defined.

2. Fig. 6 is contradictory. Why is the cost model not defined to find an optimal schedule? Why is it defined to search on tensor parallel schedules. This seems like a half-baked work. Can this be made more generic? I feel that would be more desired by the community depending on what cost metric they want to optimize.

3. If the cost function is training latency, then in fig. 6 why is the iteration time for alpha better than TAP? It seems alpha is optimal here.

4. The intuitions behind the chosen schedule are quite trivial. A more intensive case study kind of evaluation is needed



**Reviewer Expertise:**

Knowledgeable: I used to work in this area and/or I try to keep up with the literature but might not know the latest developments.